# Development of a Campus-Wide Community Service Initiative during a Pandemic

**DOI:** 10.3390/pharmacy10030047

**Published:** 2022-04-19

**Authors:** Kevan King, Hannah E. Davis, Robin Moorman-Li, Kelsey J. Cook, Nathan D. Seligson

**Affiliations:** 1Department of Pharmacotherapy and Translational Research, University of Florida, Jacksonville, FL 32209, USA; kevanking@ufl.edu (K.K.); handav93@ufl.edu (H.E.D.); moorman@cop.ufl.edu (R.M.-L.); kelsey.cook@ufl.edu (K.J.C.); 2Precision Medicine Program, Nemours Children’s Health, Jacksonville, FL 32207, USA

**Keywords:** COVID-19, volunteering, participation, involvement, organizational structure

## Abstract

Community service serves as a major aspect of pharmacy education; however, coronavirus disease 2019 (COVID-19) represented a significant disruption to student involvement. The College of Pharmacy student council, which serves as the local student government organization for the University of Florida College of Pharmacy, Jacksonville campus, developed a community service initiative to offer more consistent opportunities for students to participate in community service events, while adapting to COVID-19 restrictions. A retrospective, qualitative review of this initiative demonstrates the potential value of this model. Prior to this initiative, students relied on individual student organizations to provide service opportunities to their members. This excluded portions of the student body and led to sparse and inconsistent events, with limited variation in the types of service events available. Furthermore, de-centralized planning of service opportunities increased the difficulty of ensuring that COVID-19 safety restrictions were followed appropriately. This initiative resulted in 39 students logging over 200 service hours through nine events in the first seven months after its development. Despite the challenges presented by the COVID-19 pandemic, our centralized initiative serves as a model for improving community service involvement.

## 1. Introduction

In educating student pharmacists, colleges of pharmacy have sought additional methods to expand the didactic and experiential education of their students [1,2]. A key educational outcome for pharmacy students is the ability to provide patient- and community-centered care [3]. Community service has become a major aspect of pharmacy education [4]. Active participation in community service, among pharmacy students, has been correlated with an increase in empathy, cultural competence, and communication skills, as well as increased involvement in the community as pharmacists [4,5].

In March of 2020, the World Health Organization declared the SARS-CoV-2 viral outbreak, also referred to as the coronavirus disease 2019 (COVID-19), a pandemic. As a result of the COVID-19 pandemic and implementation of strict social distancing guidelines, colleges of pharmacies across the world transitioned classroom environments [6] and co-curricular activities onto virtual platforms [7,8,9], which resulted in significant challenges for pharmacy students [10]. The lives of students were dramatically shifted by the pandemic [11]. Many students were actively involved in providing care to patients [12], while experiencing great loss through family and friends suffering or ultimately dying from COVID-19 [13]. The COVID-19 pandemic lead to significant changes in pharmacy practice [14] and education [9,14]. The necessity of virtual classrooms led to an increase in “flipped-classroom” teaching for content delivery [15,16]. APPE rotations were modified or developed to have major online components or become completely virtual [17,18]. One area key to pharmacy education that did not adapt well to the limitations of the COVID-19 pandemic was community service. Through COVID-19, pharmacy schools saw a sharp decrease in the availability of community service opportunities [19]. Students were forced to develop new methods to serve their students and community, which adhered to COVID-19 safety protocols. Optimal methods to provide students with community service experience, while appropriately serving the community in such great need, are not well described in the literature.

At the University of Florida College of Pharmacy, Jacksonville (UF COP Jax) campus, the student council developed a centralized community service model to address deficiencies in community service opportunities caused by COVID-19. In this qualitative analysis, we describe the process we used to identify needs for community service reorganization at our college. We will provide a detailed description of the final organizational model we developed as well as preliminary results demonstrating its feasibility. This model offers significant promise in the extension of consistent, quality community service for colleges of pharmacy during COVID-19 and into the future.

## 2. Materials and Methods

### 2.1. Study Design

In this qualitative review, we retrospectively assessed the design, organization, and outcomes of a student-led, centralized community service model. The design and organization of this model was led by the authors of this study and is discussed at length here. Following the closure of community service options for students in March of 2020, the student leadership developed and implemented the described model throughout the Summer of 2020. This model was then launched in August of 2020. To identify outcomes of this new model, we collected aggregated student involvement data from 31 August 2020 through 01 April 2021. Collected aggregate data included number of events held, number of students involved, and total volunteer hours. As a result of the study being qualitative, no power analysis was conducted [20]. All data available was assessed by the full authorship team. No funding was required for this study.

### 2.2. Statistical Analysis

Descriptive statistics were used as appropriate throughout the results. No direct comparisons are made in this study; therefore, selection of an acceptable *p* value to denote statistical significance is not applicable.

## 3. Results

### 3.1. Description of the Campus Enviroment

This initiative was implemented at the UF COP Jax campus, which is one of three campuses included in the four-year Doctor of Pharmacy program of the University of Florida College of Pharmacy. The UF COP Jax campus enrolls approximately 40–50 new students each year. Students complete the Doctor of Pharmacy program entirely in Jacksonville. At the UF COP Jax campus, the student council serves as the local student government organization. The student council represents all students on the UF COP Jax campus and works collaboratively with all other student organizations on campus.

### 3.2. Development of Gators Give Back: A Campus-Wide Community Service Initiative

Prior to developing our new community service model, our campus solely relied on individual student organizations to provide service opportunities to their members. It is important to note that before this service initiative, no model for community service organization at the campus level existed. The opportunities available relied on inconsistent planning, advertisement, and community relations with individual student organizations Significant deficiencies in this existing model minimized the potential availability and impact of community service on our campus during the COVID-19 pandemic. Examples identified included limited availability of volunteer opportunities and limited access to volunteer opportunities for students who chose not to join student organizations, and inconsistency of available volunteering opportunities through the semester. Additionally, the availability of events was inconsistent. Limited variation existed in the types of service events available, due to each student organization having different mission statements and areas of interest. The presence of these concerns led to a need to rethink the model of community service at our college. Discussion with campus leaders prior to the pandemic demonstrated this need. The onset of the pandemic further exacerbated these gaps and highlighted the need for a more structured model. In order to address these deficiencies, we developed a centralized community service model which served to provide consistent and diverse community service opportunities to all students on our campus. This consistency improved the ability to ensure COVID-19 safety protocols were followed appropriately.

### 3.3. Program Structure

This initiative started with a few basic goals which included hosting one community service event per month, collaborating with one local student organization outside of the student council for each event, and partnering with as many community organizations as possible. The community service model started under the leadership of the UF COP Jax student council, which serves as the local student government organization (Figure 1). As this campus-wide model focused on including all students, regardless of professional organization membership, centralizing the model under the student council was a natural fit and allowed for maximal student involvement. The student council then appointed a dedicated student leader to serve as community service chair for the new model. This chair was responsible for developing, coordinating, marketing, and managing all service events alongside the student council president and the respective leaders of each partnered student organization. By allowing the student council community service chair to lead the general effort, we were able to relieve the individual student organizations of the burden of brainstorming, organizing, and recruiting for the events. Instead, the leaders of each partner student organization simply needed to work with the student council community service chair, in order to confirm service opportunity details and promote the event to their members. With the student organizations using their platform, in addition to the student council making each event open to the student body, event exposure was drastically increased.

The pandemic significantly influenced this initial structure. Creating events that allowed for social distancing required markedly smaller group sizes, which led to limited spots for student volunteers. As a result, we adapted a high-frequency, low-volume model, in which the frequency of events was increased, while volunteer groups remained small. This was often achieved by hosting multiple smaller events with the same community organization throughout the year. Although this initiative was designed with the focus of creating a centralized, structured community service model that spans across all student organizations. We wanted to ensure that it did not interfere with the existing initiatives created by individual student organizations. Student organizations had the ability to transition current service events to this centralized model, in partnership with the student council, or could elect to maintain independent ownership of any organization-specific service activities that they wanted to remain member-exclusive.

### 3.4. Outcomes

Aggregated outcomes were tracked for the first seven months of the implementation of this initiative. Despite the challenges presented by the COVID-19 pandemic, this initiative saw remarkable success in the first seven months after its development. In total, students logged over two hundred service hours through nine events, while working with six different community organizations. The initiative engaged a broad number of community service organizations, including Hunger Fight, Clara White Missions, Habijax Habitat for Humanity, Ronald McDonald House, Relay for Life, and Pine Castle. Additionally, the initiative hosted two donation events to provide needed items to our community and a community fundraising event for Relay for Life. This initiative also engaged with a large number of student organizations that were active on the UF COP Jax campus. Partnered student organizations included Christian Pharmacist Fellowship International, Kappa Epsilon Professional Pharmacy Fraternity, American Pharmacist Association Academy of Student Pharmacists, PediaGators, Student Society of Health-Systems Pharmacists, and Gator Pharmacy Wellness. The specific community events and corresponding partnered student organizations are listed in Table 1. This represents broad support both from local community service organizations, as well as existing student organizations on our campus.

A major limitation to studying community service in colleges of pharmacy is effectively tracking the metrics of community service involvement. When community service is de-centralized and run through individual student organizations, collecting information regarding student participation is difficult. An additional benefit of the community service initiative described here is the ability to better track changes in student participation through a centralized community service model. This allows for the further refinement of the initiative moving forward.

## 4. Discussion

Educating pharmacists requires multi-dimensional experience to bolster didactic and experiential education [1]. To meet the growing needs of our patients, pharmacy students must be able to provide patient- and community-centered care. Because of this, community service has become a major aspect of pharmacy education, leading to increased empathy, cultural competence, and communication skills. The COVID-19 pandemic necessitated the implementation of strict social distancing guidelines [6]. Colleges of pharmacy transitioned onto virtual platforms through the early stages of the COVID-19 pandemic. This resulted in significant challenges for the students, whose lives were dramatically impacted by the pandemic. The abrupt shift to a more isolated learning posed a unique challenge for students. One area key to pharmacy education that did not adapt well to the limitations of the COVID-19 pandemic was community service. Through COVID-19, pharmacy schools saw a sharp decrease in the availability of community service opportunities. Students needed to develop new, COVID-19 safe, methods to serve their fellow students and community.

In this report, we outline a novel model for a student-directed community service initiative started at the UF COP Jax campus through the student council. Students developed a new centralized community service model, which brought structure to the campus community service activities, increased student access to community service events, and created more consistent opportunities, despite the limitations imposed by the COVID-19 pandemic. The benefits of this new initiative provided many improvements to the previous model, both during the pandemic and going forward. In addition to the consistency and inclusivity this new centralized model created, there were factors of the model that were influenced by the COVID-19 pandemic that ultimately proved to be beneficial. The high-frequency, low-volume model allowed students to maintain consistent involvement with community organizations throughout the year. The increased frequency of events also allowed for a larger variety of partnerships with community organizations and varying populations, which created opportunities that appealed to many students’ individual interests, such as pediatrics, homeless populations, and community rebuilding. This provided students the ability to volunteer in ways that matched their distinct interests. Although this model resulted from the safety restrictions imposed by the pandemic, its increased the flexibility, variety, and impact to community organizations, which will prove advantageous, even in a non-pandemic future.

While this qualitative study may serve as a template for other organizations considering the structure of their community service, several limitations should be considered that affect the internal and external validity of our findings. First, the results here are from a single college of pharmacy campus. Our population, student body, and community may not be comparable to other schools of pharmacy. Tailoring of a community service model should consider local concerns by students, as well as the community. Second, the outcomes here cannot be compared to outcomes from previous years, as the lack of a centralized structure makes it impossible to track engagement with community service in the past. While this minimizes the ability to measure the comparative success of our initiative, our centralized model provides the opportunity to track and measure changes in community service engagement in the future. Furthermore, determination of the quality of community service events is significantly more difficult than quantitative assessment. Further research into determining metrics of success, both quality and quantity, is necessary. Finally, the data here represents the initial assessment of this initiative, and the results should be regarded as preliminary. Further tracking and analysis of this work is planned.

While our campus is relatively small, we believe that this model would find success in larger colleges of pharmacy, where a higher volume of students may create more inconsistencies and unpredictability, in regard to service opportunities. Based on our experience, we recommend creating a centralized model, under the local student government organization, to increase student access regardless of membership status. For a smaller campus, such as ours, this model introduced consistency and increased opportunity for our students, while allowing us to make a more profound impact on our community. We believe larger colleges of pharmacy will see similar benefits, in proportion to the size of their student body and number of events. Furthermore, a high-frequency, low-volume model, in which events are held more often in smaller groups, will allow for more variety in service programming and flexibility for student involvement. Finally, we believe that this community service initiative, while driven by the needs of the COVID-19 pandemic, would benefit colleges of pharmacy, even after all COVID-19 related restrictions are lifted.

## 5. Conclusions

Community service activities are a vital component of pharmacy education and primarily driven by student organizations. We identified inconsistencies and accessibility issues with the traditional decentralized model of organizing community service on our campus and, in response, created a centralized model, with the goal of hosting consistent and COVID-19 safe community service opportunities. This centralized model offered improved student access to community service events, while increasing the variety and flexibility of the available opportunities. Although many of the features of this model were designed to meet the needs of the community in an active pandemic, many elements of this program will continue to serve the community in a post-pandemic world.

## Figures and Tables

**Figure 1 pharmacy-10-00047-f001:**
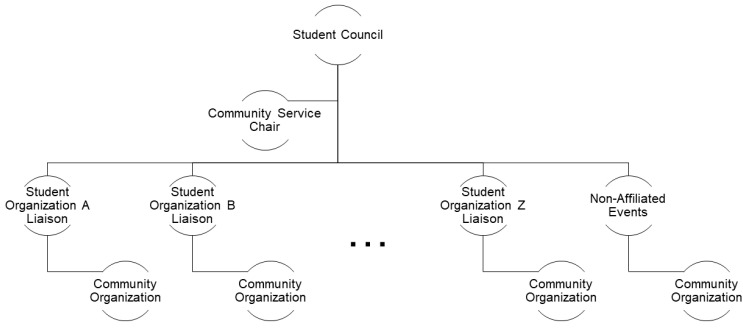
The student council community service chair serves as the lead contact with the selected student organization liaisons for volunteering activities. The community service chair coordinated with organization liaisons to organize events. The community service chair also organized events directly, which were not affiliated with any other specific student organization.

**Table 1 pharmacy-10-00047-t001:** Community service events and partner student organizations.

Community Service Event ^1^	Partnered Student Organizations
Hunger Fight	Christian Pharmacist Fellowship International
Clara White Missions	Kappa Epsilon Professional Pharmacy Fraternity
Habijax Habitat for Humanity	American Pharmacist Association Academy of Student Pharmacists
Ronald McDonald House	PediaGators
Clothing Drive	Student Society of Health-Systems Pharmacists
Relay for Life	Student Society of Health-Systems Pharmacists, Gator Pharmacy Wellness
Pine Castle	Christian Pharmacist Fellowship International

^1^ Some events were held multiple times throughout the year.

## Data Availability

Not applicable.

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
