# Peer review of "Development of a Campus-Wide Community Service Initiative during a Pandemic"

_pharmacy, 2022, doi:10.3390/pharmacy10030047_

Round 1

Reviewer 1 Report

The authors describe a new model of pharmacy students engaging in community service and events. I think this article as brief report should be shorter and therefore more interesting for the readers. Also, it would be nice to have indicators to compare decentralized and centralized model. However, I would emphasize in discussion or in conclusion the possibility of this model in the future beyond pandemic.  

I think this work is a good report, but I think it should be relatively shorter. Therefore, in my opinion, I would emphasize the difference between older and new models, with some indicators (if it is possible) and I would emphasize its relevance beyond pandemic. In addition, I would shorten the introduction part regarding pandemic, etc.

Author Response

Thank you for your review. We have revised the manuscript to be more brief. We have also provided greater explanation to emphasize that there was no existing model prior to what was implemented. This manuscript describes the first comprehensive model established on our campus. We have also expanded the discussion of the applicability of our model beyond the pandemic.

Reviewer 2 Report

Thank you for the invitation. This is an interesting piece of work highlighting the community service model in an educational institute. Since community services carry a significant role in the uplifting of society, this study carries important implications. Though authors have elaborated well on the need of the current study. However, the methods section is very limited. The authors are requested to provide information about the college of pharmacy where this model was developed and implemented. Authors may describe the ethical and administrative procedures of model development i.e., approval, funding, etc. in the method section. Since this study is harmless and avoids any ethical concerns and statistical considerations, I did not find any major flaw in this manuscript. However, the authors may improve the discussion section by incorporating the limitation of this model. I did not see a follow-up evaluation of any event reported in this study. How an event will be considered successful or not successful.--

Author Response

Thank you for your review. We have expanded the description of the college of pharmacy where this model was developed and expanded the methods section. We have highlighted the limitations of this study in the discussion and noted the future of follow-up of this study.

Reviewer 3 Report

I am grateful for the opportunity to review this manuscript. I have honestly enjoyed reading your study. I believe this study, on the Development of a Campus-Wide Community Service Initiative During a Pandemic, adds to the body of literature in pharmacy practice. Moreover, I am impressed with the up-to-date references the authors have used while writing this manuscript. Furthermore, the research is well designed and the tables are well presented.

Therefore, I believe this article should be published in its present form. Congrats to the authors.

Author Response

Thank you for your review.

Round 2

Reviewer 1 Report

Dear,

I agree with your improvements. No further comments from me.

This manuscript is a resubmission of an earlier submission. The following is a list of the peer review reports and author responses from that submission.